# Reducing Spurious Correlations in Aspect-based Sentiment Analysis with Explanation from Large Language Models

**Qianlong Wang[1,2], Keyang Ding[1,2], Bin Liang[1,2,4], Min Yang[3], Ruifeng Xu[1,2,5]***

[1]Harbin Institute of Technology, Shenzhen, China
[2]Guangdong Provincial Key Laboratory of Novel Security Intelligence Technologies
[3]Shenzhen Institute of Advanced Technology, Chinese Academy of Sciences
[4]The Chinese University of Hong Kong [5]Peng Cheng Laboratory, Shenzhen, China
qlwang15@outlook.com, keyang.ding@stu.hit.edu.cn, bin.liang@cuhk.edu.hk,
min.yang@siat.ac.cn, xuruifeng@hit.edu.cn

## Abstract

Recently, aspect-based sentiment analysis (ABSA) models have yielded promising results. However, they are susceptible to learning spurious correlations between certain words of the input text and output labels while modeling the sentiment feature of the aspect. This spurious correlation will potentially undermine the performance of ABSA models. One direct solution for this problem is to make the model see and learn an explanation of sentiment expression rather than certain words. Motivated by this, we exploit explanations for the sentiment polarity of each aspect from large language models (LLMs) to reduce spurious correlations in ABSA. First, we formulate a prompt template that wraps the sentence, an aspect, and the sentiment label. This template is utilized to prompt LLMs to generate an appropriate explanation that states the sentiment cause. Then, we propose two straightforward yet effective methods to leverage the explanation for preventing the learning of spurious correlations. We conducted extensive comparative experiments on five datasets by integrating them with some representative ABSA models. Results show that our methods can achieve performance gains and enhance the performance and generalization ability of ABSA models.

## 1 Introduction

Aspect-based sentiment analysis (ABSA) aims to identify the sentiment polarity (*e.g.*, *positive, neutral*, and *negative*) of a specified aspect in a review (Pontiki et al., 2014). For example, given a review "*great food but the service was dreadful!*" and two aspects "*food*" and "*service*", this task needs to infer their sentiment polarities "*positive*" and "*negative*", respectively.

Traditional ABSA methods primarily rely on machine learning techniques, which incorporate some handcrafted features to enhance performance,

---
* Corresponding author.

| Training Samples | Label |
|---|---|
| company provides UPS [***shipping***], fast, great! | *positive* |
| the [***Final Cut Pro***] on this laptop is so fast and easy. | *positive* |
| super fast [***processor***] and really nice graphics card. | *positive* |
| **Testing Sample** | |
| the [***battery life***] was faster than expected. | *negative* |
| *(prediction: positive)* | |

Figure 1: Examples of the spurious correlation between context words "*fast*" and label "*positive*" in the training samples. This spurious correlation does not hold for the testing sample. Aspect terms are marked in parentheses.

such as linguistic features (Negi and Buitelaar, 2014). However, feature engineering could be a time-consuming process, requiring significant effort and expertise. To solve this dilemma, deep learning solutions are utilized to address ABSA due to their powerful contextual feature modeling capability. From conventional neural networks (Ruder et al., 2016; Xue and Li, 2018; Ma et al., 2018) to attention mechanisms (Tang et al., 2016a; Li et al., 2018a; Gu et al., 2018; Fan et al., 2018), these solutions focus on modeling the dependency relationship between an aspect and its corresponding opinion expressions. With the emergence of fine-tuning paradigm, the attention mechanism armed with pre-trained language models (PLMs) (Devlin et al., 2019; Song et al., 2019; Wang et al., 2020; Tian et al., 2021; Nazir et al., 2022; Zhang et al., 2022) further strengthens the connection between the aspect and its context.

Despite their satisfactory results, most neural network methods may indulge in learning statistically spurious correlations while modeling the sentiment feature of aspect on the context. Here, *spurious correlation* (Wang and Culotta, 2020; Wang et al., 2022b) refers to the dependence of the model on

certain words of the input text without a deeper understanding of the contextual semantics, which has a *know-it-when-you-see-it* character. Taking the example in Figure 1 for illustration, the opinion word "*fast*" expresses different sentiment polarities of the aspect terms in distinct contexts. Here, 92% of aspect sentiment is "*positive*" in the training samples when counting the proportion of aspect sentiment polarity that co-occurs with "*fast*". Due to this unbalanced distribution, in the training phase, the neural models assume that there is a strong correlation between "*fast*" and "*positive*", especially for short texts. Consequently, when faced with a testing sample containing a derivative "*faster*", the trained models will predict the incorrect sentiment label "*positive*" based on this spurious correlation learned superficially before. Thus, most neural models may encounter difficulty in navigating spurious correlations because of a shallow understanding. Besides, they lack the capacity to self-correct, resulting in undermining their effectiveness and generalization.

One straightforward solution to alleviate the spurious correlation problem is to make models attend to an explanation of sentiment expression rather than certain words in the context. Here, *explanation* refers to the reasons for the sentiment polarity of aspect term obtained by deeply understanding the contextual semantics. However, for each training sample, it is a tricky problem to derive the sentiment explanation given the aspect and its sentiment. Recently, large language models (LLMs) (Brown et al., 2020) have achieved remarkable success in a wide range of NLP capabilities, including generation and contextual understanding. In addition, they are knowledgeable due to the substantial linguistic (Liu et al., 2019) and factual world knowledge learned. Thus, LLMs can be exploited to generate an explanation toward the sentiment of aspect through prompt-driven contextual understanding (Bian et al., 2023). Taking the second training sample in Figure 1 as an example, LLMs can yield an explanation, "*The sentiment towards 'Final Cut Pro' is positive because the speaker praises its efficiency and user-friendliness on the laptop, indicating satisfaction and favorable feelings about the software.*".

Inspired by this, we leverage explanations from LLMs to reduce spurious correlations in ABSA. Specifically, we first design a prompt containing an aspect term and its sentiment to induce LLMs

to provide a relevant explanation according to the context. In this way, the output explanation can provide the reason for sentiment and may contain some external knowledge thanks to the powerful capabilities of LLMs. Then, we propose two methods to employ this explanation to improve the effectiveness and generalization of ABSA models. One is the *augmentation-based method*, which directly treats these explanations containing the aspect term as training samples. We mix these explanations with the original training samples to train a more robust ABSA model. This method can not only relieve the statistical bias in original samples but also learn a range of sample patterns. The other is the *distillation-based method*, whose basic idea is to distill the knowledge embedded in the explanation into a student ABSA model. By the distillation loss, the student ABSA model can mimic the two behaviors of the teacher, *i.e.*, sentiment representation and output logit. In this way, the explanation can guide the learning of ABSA models and prevent them from over-focusing on spurious correlations.

In summary, our contributions are as follows:

- To our knowledge, we are the first to induce LLMs to generate an explanation for the aspect's sentiment and use it to reduce spurious correlations in the ABSA task.

- We devise two straightforward methods for utilizing this explanation, which can be integrated into most mainstream baselines.

- We conduct extensive experiments on five benchmark datasets, showing that baselines armed with the proposed methods can achieve better performance on inference and generalization.

## 2 Related Work

**Aspect-based Sentiment Analysis.** ABSA aims to identify the sentiment polarity of each aspect mentioned in the text. To solve this task, various neural networks with the attention mechanism are utilized to find the semantic relation of an aspect and its context for capturing the corresponding opinion expression (Tang et al., 2016b; Ma et al., 2017; Li et al., 2018c; Fan et al., 2018; Tan et al., 2019). For instance, Fan et al. (2018) exploited a multi-grained attention mechanism to capture the word-level interaction between the aspect and its relevant context. The idea behind the attention

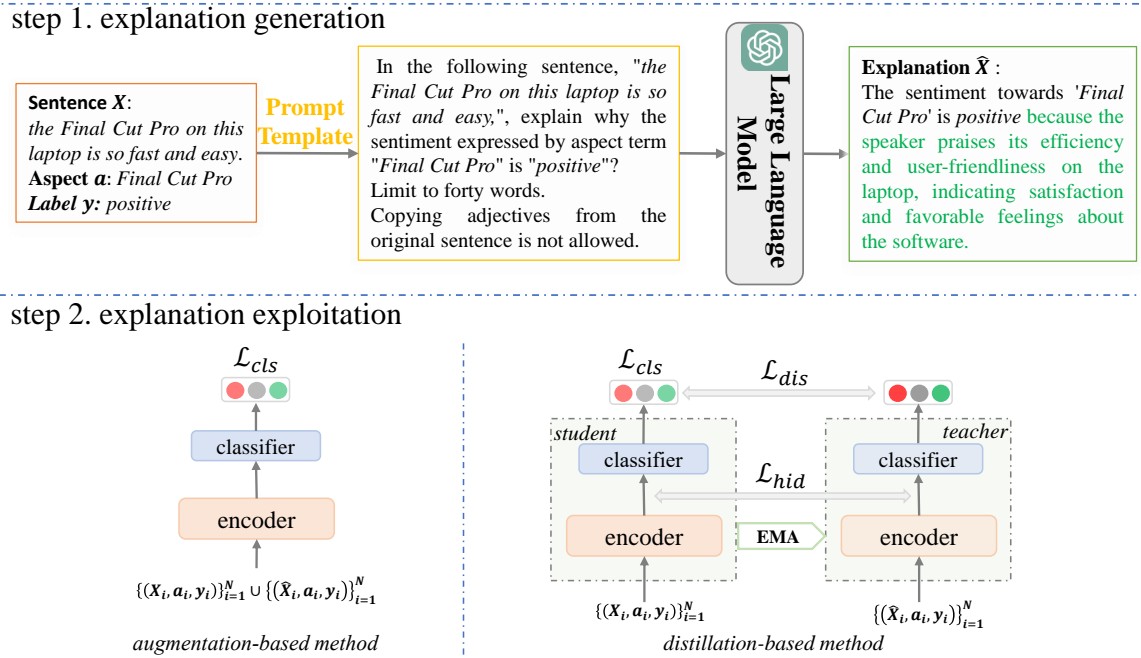

Figure 2: The overview of the proposed framework. This framework consists of two steps: explanation generation and explanation exploitation.

mechanism is to focus on the context related to the aspect and shield the irrelevant context. To further pursue this idea, some studies (Song et al., 2019; Li et al., 2020; Yan et al., 2021; Wang et al., 2022b,a) applied pre-trained models (PLMs) such as BERT (Devlin et al., 2019) to model the semantic relationship between the given aspect and its context. The internal multi-head self-attention mechanism in PLMs is more efficient than conventional attention techniques (Vaswani et al., 2017). As a result, these studies consistently delivered better results.

Another research trend is to leverage syntactic knowledge from syntactic trees to handle ABSA. This syntactic knowledge helps to establish connections between the aspect and opinion words and learn syntax-aware feature representations of the aspect (He et al., 2018; Sun et al., 2019; Phan and Ogunbona, 2020; Wang et al., 2020; Tian et al., 2021; Liang et al., 2022). The core idea of these studies is to transform a constructed syntax dependency tree into a graph for posing greater attention to important words.

Although these methods obtained promising results by modeling semantic relationships between aspects and contexts, they are inevitably plagued by statistical spurious correlations. Unlike them, in this paper, we aim to reduce spurious correlations with explanations from LLMs. These explanations can serve to guide ABSA models not to

focus on certain words in the context to prevent being trapped in the spurious correlation trap.

**Large Language Models.** With the advent of GPT-3 (Brown et al., 2020), LLMs break into the limelight and draw enormous attention. They typically feature a vast array of model parameters and undergo training on immensely large volumes of raw data. By learning from data, they memorize and understand vast amounts of knowledge (Li et al., 2022) and learn to reason (Wei et al., 2022). Knowledge and reason are crucial for building a satisfactory NLP system that can understand and generate human-like language. Consequently, LLMs like ChatGPT can achieve substantial performance improvements in a wide range of NLP tasks, including inference and dialogue, by profoundly comprehending the contextual semantics. Inspired by this, for the sentiment polarity of each aspect, we here apply LLMs to generate an explanation to explain its causes.

## 3 Our Approach

### 3.1 Problem Definition

Given an ABSA training set, each sample consists of a sentence $X$, an aspect $a$, and a sentiment label $y$. Here, the aspect is a sub-sequence token in the sentence. ABSA aims to learn a sentiment classifier that can precisely predict a sentiment polarity $y \in$

{*positive, negative, neural*} for each aspect term according to the semantics of the sentence.[1]

## 3.2 Overview

As shown in Figure 2, our framework consists of two steps. The first step is *explanation generation*. Here, for each training sample, we use a prompt template to encapsulate the sentence, the aspect, and its sentiment to drive the LLMs to generate an explanation to indicate the corresponding sentiment cause. The second step is *explanation exploitation*. Here, we propose two simple yet effective methods to exploit explanations for alleviating the spurious correlations in ABSA.

## 3.3 Explanation Generation

Spurious correlations are common in current ABSA models, particularly in cases of over-parameterization or insufficient training data (Sagawa et al., 2020). The fundamental reason is that these models might learn statistical correlations between superficial textual cues and sentiment labels rather than achieving a profound comprehension of contextual semantics. Consequently, this problem will hurt the performance and generality of the ABSA classifier.

In this work, we try to reduce spurious correlations in ABSA using *explanation*. To achieve this, we expect an explanation to have two functions: (*i*) motivating ABSA models to infer the aspect's sentiment by understanding the context rather than some surface words. (*ii*) providing additional knowledge as background information for better contextual understanding, especially for short texts; Recently, LLMs such as ChatGPT have exhibited incredible contextual understanding and knowledge inference on a wide range of NLP (Wei et al., 2022). It inspires us to leverage LLMs to generate an explanation for the aspect's sentiment in each training sample, which has not been explored in the literature. To this end, we design a prompt template to trigger the understanding and inference ability of LLMs, which wraps the sentence $X$, an aspect $a$, and its sentiment $y$:

> *In the following sentence X, explain why*
> *the sentiment expressed by aspect term*
> ***a** is **y**. Limit to forty words. Copying*

*adjectives from the original sentence is not allowed.*

We can see that this prompt consists of three components: task description, training sample, and output limitation. They describe the task precisely and form a good output guide, which helps to enhance the generation performance. As shown in the example in Figure 2, LLM is tasked with generating a friendly explanation $\widehat{X}$ for the aspect sentiment. This explanation not only explains why the sentiment occurs based on contextual semantics (*i.e., "user-friendliness"*) but also includes some background knowledge (*i.e., "software"*).

## 3.4 Explanation Exploitation

The explanation generated by the LLM provides us with a comprehensive view of the original text from the perspective of contextual semantics and background knowledge. Furthermore, the explanation does not have high-frequency adjectives (*e.g., fast*) due to the limitation in the prompt, which further provides a sufficient condition to mitigate statistical spurious correlations. Thus, we can use them to aid the learning of ABSA models and improve the performance of the model. Here, we present two straightforward and model-agnostic methods to achieve this.

**Augmentation-based Method.** In a sense, the explanation can be considered as a paraphrase of the original sentence, which has the same semantic meaning and label but a different description. This different description not only facilitates the alleviation of statistical bias in the original sentences but also diversifies the expression of the same sentiment. Thus, mixing the original training data $\{(X_i, a_i, y_i)\}_{i=1}^{N}$ with their explanations $\{(\widehat{X}_i, a_i, y_i)\}_{i=1}^{N}$ can allow for training a more robust ABSA classifier:

$$\mathcal{L}_{cls} = -\frac{1}{2N} \sum_{i=1}^{2N} \text{CE}(y, P(X_i', a))  \quad (1)$$

where $P(X', a)$ is the predictive probability distribution of sentiment; $X'$ can be either $X$ or $\widehat{X}$; $CE$ denotes the cross-entropy loss function.

The explanation is more effective than conventional data augmentation methods (Wei and Zou, 2019) as it interprets the contextual semantics and contains some knowledge.

---

[1]For sentences with multiple aspects, we treat other non-targeted aspects as normal context tokens when focusing on the target aspect. In other words, a sentence will be processed multiple times, which is equal to the number of aspects it contains.

**Distillation-based Method.** Direct mixing cannot align the original sentence with the corresponding explanation, which results in trouble providing customized guided learning. To this end, we use a guidance strategy to encourage ABSA models to reduce spurious correlations in fitting each sample. This strategy can be viewed as a knowledge distillation (Hinton et al., 2015), which aims to leverage the teacher to guide the student's training with the help of explanations.[2] To achieve this guidance, we here make the student model mimic two behaviors of the teacher one via the following two losses:

$$\mathcal{L}_{dis} = \frac{1}{N} \sum_{i=1}^{N} \text{KL}(g_s(X_i', a), g_t(X_i', a)) \quad (2)$$

$$\mathcal{L}_{hid} = \frac{1}{N} \sum_{i=1}^{N} \text{MSE}(h_{X_i'}^s, h_{X_i'}^t) \quad (3)$$

where $g_s(X_i', a)$ and $g_t(X_i', a)$ ($h_{X_i'}^s$ and $h_{X_i'}^t$) refer to the logits (hidden states), which come from student and teacher networks, respectively; $KL$ denotes the Kullback-Leibler divergence loss function; $MSE$ denotes the mean squared error loss function. By two losses, the explanation is utilized to facilitate the learning process of ABSA models and mitigate overly concentrated on shortcut features.

To yield better guidance, the teacher network tracks an exponential moving average (Tarvainen and Valpola, 2017) of the student network weights. At the time step $t$, the parameters of the teacher $\theta$ are updated as follows:

$$\theta_t = \lambda \cdot \theta_{t-1} + (1 - \lambda) \cdot \phi_t \quad (4)$$

where $\phi_t$ represents all parameters in the student network at time step $t$; $\lambda$ is a smoothing coefficient. With this moving average, the teacher's output is more reliable.

### 3.5 Training and Testing

For the augmentation-based method, we train the parameters of the ABSA model directly by optimizing $\mathcal{L}_{cls}$ (Eq. 1). For the distillation-based method, we update parameters of the student model by optimizing the sum of $\mathcal{L}_{cls}$ (Eq. 1), $\mathcal{L}_{dis}$ (Eq. 2), and $\mathcal{L}_{hid}$ (Eq. 3). In the test phase, the sentence and aspect are fed into the student network to predict the label.

---

[2]In this work, the teacher model and the student model have the same framework.

## 4 Experiment

### 4.1 Datasets and Settings

**Datasets.** We use five benchmark datasets to evaluate the proposed methods: Lap14 and Rest14 from Pontiki et al. (2014), Rest15 from Pontiki et al. (2015), Rest16 from Pontiki et al. (2016), and MAMS from Jiang et al. (2019). All datasets only involve three sentiment labels, *positive, neutral*, and *negative*. Each sample in these datasets is annotated with aspects and their corresponding sentiment polarities. Here, we adopt the official data splits as done in the original papers. The basic statistics are shown in Table 1.

| Dataset | | #Pos | #Neu | #Neg | Total |
|---|---|---|---|---|---|
| Lap14 | train | 994 | 464 | 870 | 2,328 |
| | test | 341 | 169 | 128 | 638 |
| Rest14 | train | 2,164 | 637 | 807 | 3,608 |
| | test | 728 | 196 | 196 | 1,120 |
| Rest15 | train | 912 | 36 | 256 | 1,204 |
| | test | 326 | 34 | 182 | 542 |
| Rest16 | train | 1,240 | 69 | 439 | 1,748 |
| | test | 469 | 30 | 117 | 616 |
| MAMS | train | 3,380 | 5,042 | 2,764 | 11,186 |
| | dev | 403 | 604 | 325 | 1,332 |
| | test | 400 | 607 | 329 | 1,336 |

Table 1: The detailed statistics of datasets.

**Settings.** If not otherwise specified, we use Chat-GPT and the pre-trained uncased BERT-base[3] as LLM and encoder in the framework[4], respectively. For the classifier, the weight matrix is randomly initialized by a uniform distribution. To avoid over-fitting, we apply the dropout (Srivastava et al., 2014) with a probability of 0.1. Besides, we also replace the label words (i.e., *positive, neutral*, and *negative*) in the explanation with [MASK] token. We employ the AdamW optimizer to optimize parameters. The epoch, batch size, learning rate, and smoothing coefficient are set to 8, 24, 3e-5, and 0.95, respectively. We limit the maximum length of the token sequence to 256.

We run the experiments five times with random initialization and report the averaged results. The accuracy (**Acc.**) and macro-averaged F1 (**F1**) scores are used as the evaluation metric.

### 4.2 Baselines

To evaluate the effectiveness and generalization of the proposed methods, we integrate them with

---

[3]https://github.com/google-research/bert
[4]The proposed framework only loads the LLM for inference without involving training.

| Datasets / Models | Lap14 Acc. | Lap14 F1 | Rest14 Acc. | Rest14 F1 | Rest15 Acc. | Rest15 F1 | Rest16 Acc. | Rest16 F1 | MAMS Acc. | MAMS F1 |
|---|---|---|---|---|---|---|---|---|---|---|
| *Large Language Models* | | | | | | | | | | |
| MOSS (zero-shot) | 68.34 | 51.18 | 75.89 | 52.22 | 81.55 | 58.40 | 85.06 | 58.35 | 41.92 | 36.21 |
| MOSS (few-shot) | 70.85 | 61.21 | 77.59 | 61.81 | 82.29 | 59.04 | 87.18 | 64.04 | 43.71 | 42.43 |
| ChatGPT (zero-shot) | 78.64 | 70.70 | 80.39 | 71.33 | 77.01 | 63.21 | 83.39 | 68.75 | 63.67 | 64.08 |
| ChatGPT (few-shot) | 78.90 | 75.61 | 83.74 | 77.08 | 82.80 | 71.31 | 84.83 | 69.90 | 60.18 | 60.57 |
| *the PLMs-based Models* | | | | | | | | | | |
| BERT | 75.80 | 71.67 | 82.59 | 74.10 | 80.97 | 63.52 | 88.47 | 71.16 | 82.68 | 82.37 |
| + augmentation | 77.74 | 73.11 | 84.29 | 76.07 | 82.03 | 65.80 | 89.96 | 73.06 | **84.51** | **84.11** |
| + distillation | **78.68** | **75.19** | **84.66** | **76.18** | **82.63** | **65.97** | **90.12** | **73.69** | 83.68 | 83.38 |
| BERT+PT | 77.59 | 73.14 | 84.16 | 76.62 | 82.83 | 65.81 | 91.73 | 73.93 | 83.48 | 83.10 |
| + augmentation | **77.90** | **74.66** | 85.71 | 78.37 | 83.50 | 67.48 | **92.75** | **75.48** | 83.86 | 83.73 |
| + distillation | 77.54 | 73.71 | **85.96** | **78.93** | **84.14** | **68.05** | 92.02 | 74.45 | **84.43** | **83.92** |
| *the Attention-based Models* | | | | | | | | | | |
| TNet | 76.63 | 72.00 | 82.68 | 74.33 | 81.55 | 64.99 | 88.80 | 71.41 | 82.84 | 82.42 |
| + augmentation | 77.82 | 73.24 | 84.30 | 75.96 | 82.48 | **66.28** | 90.12 | 73.22 | 83.73 | 83.15 |
| + distillation | **78.04** | **73.32** | **84.75** | **76.68** | **82.97** | 66.17 | **90.30** | **73.81** | **83.98** | **83.42** |
| AEN | 77.27 | 72.68 | 83.66 | 76.02 | 82.00 | 67.27 | 89.45 | 73.61 | 82.86 | 82.58 |
| + augmentation | 78.37 | **74.25** | **85.54** | **79.58** | **83.03** | **69.85** | 90.42 | 75.04 | **83.34** | **83.03** |
| + distillation | **78.40** | 73.96 | 84.20 | 77.20 | 82.47 | 67.84 | **91.05** | **75.45** | 83.19 | 82.70 |
| *the Graph-based Models* | | | | | | | | | | |
| RGAT | 77.45 | 72.70 | 86.02 | 80.74 | 81.80 | 68.21 | 89.51 | 75.81 | 82.93 | 82.75 |
| + augmentation | **80.31** | **75.73** | **87.45** | **82.41** | **83.86** | **70.42** | **91.61** | **77.44** | **84.61** | **84.03** |
| + distillation | 78.11 | 74.06 | 86.30 | 81.12 | 82.55 | 69.29 | 90.47 | 77.05 | 83.78 | 83.18 |
| DualGCN | 80.62 | 74.67 | 85.20 | 80.16 | 82.33 | 68.12 | 90.91 | 77.86 | 83.83 | 83.47 |
| + augmentation | **81.56** | **75.92** | 86.18 | 80.50 | **83.98** | **70.86** | 91.12 | 77.97 | 84.28 | 83.94 |
| + distillation | 81.22 | 75.43 | **86.37** | **80.63** | 82.69 | 69.43 | **91.45** | **78.12** | **84.68** | **84.23** |

Table 2: Main experimental results (%) on five ABSA datasets. The zero-shot and few-shot indicate that LLM uses 0 and 4 demonstrations in the in-context learning paradigm (Dong et al., 2022), respectively. All the results are the average of five seeds. For each baseline, the best scores are in bold.

some representative ABSA models and compare performance. These ABSA models could be categorized into three groups. (1) ***the PLMs-based models***, which includes BERT (Devlin et al., 2019) and BERT-PT (Xu et al., 2019). (2) ***the attention-based models***, which includes TNet (Li et al., 2018b) and AEN (Song et al., 2019). (3) ***the graph-based models***, which includes RGAT (Wang et al., 2020) and DualGCN (Li et al., 2021).

In addition to the ABSA models mentioned above, we also introduce two LLMs (MOSS[5] and ChatGPT[6]) as strong competitors.

### 4.3 Main Results

Table 2 shows the experimental results of ABSA models on five datasets. We can draw the following conclusions from this table:

**First**, ABSA models equipped with our methods (*i.e.*, + *augmentation* and + *distillation*) achieve better performance than peer competitors on both accuracy and F1. Among them, the bigger improvements in accuracy and F1 are 2.88% (BERT+distillation on the Lap14) and 3.56% (AEN+augmentation on the Rest14), respectively. These improvements show that (*i*) the proposed explanation can effectively mitigate spurious correlations, and (*ii*) our methods can be seamlessly compensated to existing ABSA models.

**Second**, the graph-based ABSA models perform better than the attention-based ones. For example, DualGCN improves performance by 3.99% in accuracy and 3.67% in F1 over TNet on the Lap14. Although the graph-based models have obtained satisfactory results, we can observe a boost of 0.28~2.86% in accuracy and 0.38~3.03% in F1 when integrated with our methods. It indicates that while exploiting the syntactic knowledge connecting aspects and opinion words to improve performance, they may still model shortcut features

[5] The snapshot version and parameters of the MOSS are MOSS-moon-003-sft and 16B, respectively. Please refer to https://moss.fastnlp.top/
[6] The snapshot version and parameters of the ChatGPT are text-davinci-003 and 175B, respectively. Please refer to https://openai.com/blog/chatgpt

| Datasets | Lap14 | | Rest14 | | Rest15 | | Rest16 | | MAMS | |
|---|---|---|---|---|---|---|---|---|---|---|
| Models | Acc. | F1 | Acc. | F1 | Acc. | F1 | Acc. | F1 | Acc. | F1 |
| BERT | 75.80 | 71.67 | 82.59 | 74.10 | 80.97 | 63.52 | 88.47 | 71.16 | 82.68 | 82.37 |
| + SWAP | 77.03 | 72.33 | 83.24 | 75.15 | 81.32 | 64.68 | 89.58 | 72.42 | 82.76 | 83.00 |
| + ADD | 77.21 | 72.46 | 83.80 | 75.71 | 81.55 | 64.81 | 89.10 | 72.59 | 83.31 | 82.85 |
| + DELETE | 76.84 | 72.56 | 82.95 | 73.22 | 80.63 | 63.02 | 89.26 | 71.57 | 83.03 | 82.88 |
| + MASK | 76.65 | 72.64 | 83.57 | 74.70 | 79.15 | 60.50 | 89.12 | 71.58 | 83.06 | 82.80 |
| + TRANSLATION | 77.27 | 72.80 | 84.08 | 74.90 | 81.52 | 65.18 | 89.49 | 72.75 | 82.91 | 83.07 |
| + augmentation | 77.74 | 73.11 | 84.29 | 76.07 | 82.03 | 65.80 | 89.96 | 73.06 | **84.51** | **84.11** |
| + distillation | **78.68** | **75.19** | **84.66** | **76.18** | **82.63** | **65.97** | **90.12** | **73.69** | 83.68 | 83.38 |

Table 3: Comparison of our methods with five popular data augmentation baselines on the ABSA task. **SWAP**: randomly swap two tokens; **ADD**: randomly insert some sampled tokens; **DELETE**: randomly remove some tokens; **MASK**: first replace some tokens with [MASK] token and then use BERT to complete the mask language task (Devlin et al., 2019); **TRANSLATION** (Sennrich et al., 2016): first translate the text into Chinese and then translate the output into English. The best scores are in bold.

because of a shallow understanding of some words.

**Third**, LLMs can yield impressive results using few demonstrations. Compared with PLMs, they are scaling up in depth and width. It causes them to become increasingly computationally and storage-intensive, making deployment difficult. This is why we leverage explanations from LLMs to reduce the spurious correlations in ABSA rather than using them directly to solve it.

**Fourth**, we find that the augmentation-based method and distillation-based one could not tell who wins and who loses. Each has its own advantages and merits. For example, although the distillation-based method yields higher results than the augmentation-based method in some cases, the latter is superior with respect to efficiency. In addition, the augmentation-based method is more applicable in different ABSA models. Therefore, we will subsequently focus more on the augmentation-based method for future research.

| | Lap14 | Rest14 |
|---|---|---|
| BERT | 14.8 | 14.5 |
| + augmentation | **12.7** | 11.6 |
| + distillation | 13.0 | **11.4** |
| AEN | 16.3 | 16.4 |
| + augmentation | **15.2** | **14.8** |
| + distillation | 15.3 | 15.0 |

Table 4: The proportion (%) of spurious correlations present in the dataset. The lower the better.

## 5 Discussion

**Percentage of Spurious Correlations in the Dataset.** In this work, spurious correlation (Wang et al., 2022b) refers to the dependence of the model on certain words in the input text without a deeper understanding of the contextual semantics. A question naturally arises how much of the correlation

actually is in used datasets? To answer this question, we conduct a simple probe experiment on the Aspect Robustness Test Set (ARTS) (Xing et al., 2020). ARTS enrich the initial test sets from Lap14 and Rest14 by employing three adversarial strategies[7]. Here, *if a model predicts the same sentiment labels for an original sample as well as its adversarial samples (their true labels are different), we will assume that there is a spurious correlation between certain contextual words and the predicted label for this original sample*. In other words, the predicted label does not change with the contextual semantics because the model only focuses on certain words. Based on this assumption, we count the percentage of original samples in the test set that contain spurious correlations.[8] According to Table 4, we can see that spurious correlations do exist in the Lap14 and Rest14 datasets. Moreover, we can observe that the proposed methods reduce the percentage of original samples containing spurious correlations. This may suggest that the generated explanations can alleviate the spurious correlations problem in the ABSA task.

**Comparison with Data Augmentation Baselines.** This work aims to exploit the explanation from LLMs to reduce spurious correlations, which could be viewed as an augmented instance of the original sentence. To evaluate its effectiveness, we compare the proposed methods with five data augmentation

---

[7]They are (1) reversing the original sentiment of the targeted aspect; (2) reversing the sentiment of the non-targeted aspects; and (3) generating more non-targeted aspects with opposite sentiment polarities from the targeted aspect.

[8]It is worth reminding that this percentage is not the actual percentage of spurious correlations in the dataset, which is only an estimate under this assumption.

| Datasets | $\mathbb{L} \Rightarrow \mathbb{R}$ | | $\mathbb{R} \Rightarrow \mathbb{L}$ | |
| Models | Acc. | F1 | Acc. | F1 |
|---|---|---|---|---|
| BERT | 77.02 | 63.83 | 73.82 | 68.74 |
| + augmentation | 78.93 | **66.33** | **76.33** | **72.02** |
| + distillation | **79.11** | 64.93 | 75.55 | 70.99 |
| BERT+PT | 78.82 | 70.74 | 74.55 | 68.62 |
| + augmentation | 81.25 | 71.42 | **75.55** | **69.91** |
| + distillation | **82.68** | **75.37** | 73.20 | 66.53 |

Table 5: The generalization results of ABSA models. $\mathbb{L} \Rightarrow \mathbb{R}$ (or $\mathbb{R} \Rightarrow \mathbb{L}$) refer to that the model is trained on Lap14 (or Rest14) training data and then tested on Rest14 (or Lap14) test data. The best scores for each baseline are in bold.

baselines.[9] Table 3 reports the experimental results. It can be seen that our methods perform better than all baselines, achieving the biggest improvements of 3.48% and 5.47% in accuracy and F1, respectively. This shows that the explanation from LLMs is more effective because of including not only a contextual understanding of sentiments but also external knowledge. Besides, we find that these augmentation baselines often consistently improve the performance of BERT, showing that modifying the original text may bring gains, despite the noise.

**Generalization Analysis.** We evaluate the proposed methods in the cross-domain scenario to check their effectiveness on generalizability. The experimental results are presented in Table 5. We can observe that: (1) Our methods significantly enhance the generalization ability of the peer baselines by a substantial margin. We attribute it to the explanations of aspect sentiment that can reduce the spurious correlations in ABSA. (2) Overall, the *augmentation* method is more effective than the *distillation* one. A possible reason for this is that explanations containing external knowledge are directly involved in the training, allowing the learning of better transfer features.

**Effectiveness in Low-Resource Scenario.** Here, we carry out an experiment to observe the performance improvements achieved by our proposed methods in low-resource settings. To this end, we vary the percentage of training data from 10% to 100% in increments of 10% and depict results in Figure 3. We can see that: (1) Overall, being armed with our methods can improve the performance of BERT. It shows that introducing an explanation for sentiment is useful in low-resource scenarios. (2) The performance gradually improves as the per-

[9]We add augmented versions directly to the existing set. Thus, the training set size will be doubled after this operation.

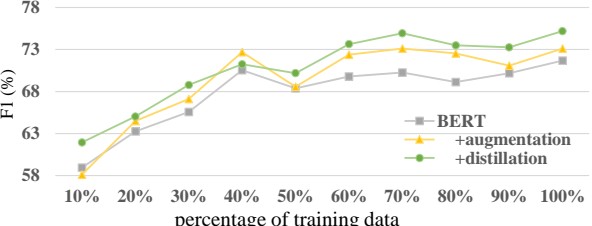

(a) On Lap14 dataset.

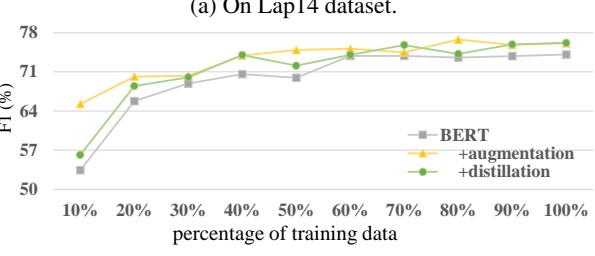

(b) On Rest14 dataset.

Figure 3: Experiments in low-resource scenarios. We limit the percentage of training data when fine-tuning.

centage increases before the training size surpasses 50%, indicating that the more training data, the better the model is trained. Nevertheless, upon surpassing this point, the performance fluctuates moderately.

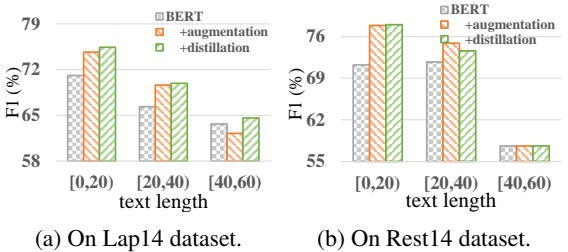

(a) On Lap14 dataset.     (b) On Rest14 dataset.

Figure 4: Length-wise performance comparison. Here, [20, 40) indicates that the token number of the sample satisfies the condition of greater than or equal to 20 and less than 40. The other meanings are similar.

**Length-wise Performance Analysis.** Spurious correlations are prone to occur when predicting the short text as the model tends to resort to the learned statistical bias facing a low-informative context. Here, we test length-wise performance to reveal the noticeable advantages of the proposed methods on short texts. Figure 4 provides the test results. We can see that the proposed methods significantly improve the performance of BERT, especially on short texts. It shows that the explanation provided by LLMs for the training samples motivates the ABSA model to be trained effectively, thus allowing for a better contextual understanding of short texts at testing.

**The Appendix has more discussion and analysis**, *i.e.*, Quality of the Automatically-Generated Explanations, Effect of Different Prompt Templates on Performance, and Error Analysis.

## 6 Conclusion

In this paper, we introduce an effective two-step framework to mitigate spurious correlations in ABSA. First, we formulate a prompt template to induce LLMs to generate an appropriate explanation that states the sentiment cause. Subsequently, we propose two straightforward methods that utilize the generated explanation to prevent the assimilation of spurious correlations. Our comprehensive experiments on five ABSA datasets show that baselines armed with our methods outperform peers in prediction performance and generalization.

## Limitations

In this section, we list two limitations to understand this work more comprehensively:

1. *The prompt template designed in this work consists of three components: the task description, the training sample, and the output limitation.* Generally speaking, a prompt-rich template allows LLMs to generate more helpful explanations about sentiment and richer relevant external knowledge. In this work, we did not design a prompt-rich template because this manual design process is time-consuming and cumbersome. In addition, designing a complex and information-rich prompt template is not the research focus of this work.

2. *We leverage explanations to reduce spurious correlations in the ABSA task.* In this work, we generate an explanation for the sentiment label in each training sample, which subsequently participates in the model training process. Although spurious correlations are caused by statistical bias during training, not all training samples bring bias interference to the model. Therefore, the participation of all explanations in model training is an extensive operation, which somehow results in a waste of training resources. How to identify whether a training sample potentially brings spurious correlation interference can be a direction for subsequent research.

## Acknowledgements

This work was partially supported by the National Natural Science Foundation of China (62006062, 62176076), Natural Science Foundation of Guangdong 2023A1515012922, Shenzhen Foundational Research Funding JCYJ20210324115614039 and JCYJ20220818102415032, Guangdong Provincial Key Laboratory of Novel Security Intelligence Technologies 2022B1212010005.

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

# 7 Appendix

**Quality of the Automatically-Generated Explanations.** Large language models may generate less accurate or even irrelevant explanations for the sentiment of the aspect. Such explanations may have negative effects if they exist and are involved in training. In this work, we only randomly selected nearly fifty explanations for hand-checking. We find that the quality of the generated explanations is accurate and comprehensive (see examples in Table 6). In addition, We observe that these explanations are richly expressive (*e.g., "not reasonable or affordable"*).

**Effect of Different Prompt Templates on Performance.** In this work, the prompt template is designed based on two points: (1) including a task description (*i.e., "explain the sentiment expressed by aspect term"*), which is used to trigger the LLM's ability to understand the task; (2) including the output limitation (*i.e., "Limit to forty words. Copying adjectives from the original sentence is not allowed."*), which is used as an output guide and prevents LLM from rephrasing the sentiment expression. Here, to explore the effect of different prompt templates on performance, we try other well-designed templates and perform comparison experiments. These prompt templates are described in detail below:

- **Prompt 1** (used in this paper): In the following sentence $X$, explain why the sentiment expressed by aspect term $a$ is $y$. Limit to forty words. Copying adjectives from the original sentence is not allowed.

- **Prompt 2**: In the following sentence $X$, explain why the sentiment expressed by aspect term $a$ is $y$.

- **Prompt 3**: In the following sentence $X$, the sentiment expressed by aspect term $a$ is $y$. Based on this prompt, explain its reasoning. Limit to forty words. Copying adjectives from the original sentence is not allowed.

Here, for simplicity, we choose BERT as the baseline model and BERT+augmentation as our method. Table 8 presents the experimental results. From this table, we can find that although the template used in this paper yields the best results, there is not much difference between the scores of the different templates (see Prompt 1 *vs.* Prompt 3). Moreover, we observe that imposing an output constraint in the template favors the performance (see Prompt 1 *vs.* Prompt 2).

**Error Analysis.** An error analysis can provide readers with a deeper understanding of whether our

| Dataset | Sample | Aspect | Sentiment | Explanation |
|---|---|---|---|---|
| Lap14 | The machine is slow to boot up and occasionally crashes completely. | boot up | negative | The phrase "boot up" implies that *the machine is taking a long time to start*, which is a negative experience. |
| Rest14 | Even though its good seafood, the prices are too high. | prices | negative | The prices are too high, which implies that *they are not reasonable or affordable.* |
| MAMS | USC has a cold smoker and smoked the avocado in order to make the dish more complex. | dish | negative | The dish is negative *because the avocado was smoked, which could make the dish taste unpleasant or unappetizing.* |

Table 6: Examples of generating explanations. We can see that the explanation elucidates the reason (*italicized content*) for the sentiment of the aspect, grounded in contextual semantics.

| Sample | BERT | BERT+augmentation |
|---|---|---|
| The $[staff]_{negative}$ should be a bit more friendly. | positive(✗) | negative(✓) |
| I needed a laptop with big $[storage]_{neutral}$, a nice screen and fast so I can Photoshop without any problem. | positive(✗) | positive(✗) |

Table 7: A simple error analysis. Aspect terms are marked in parentheses, and the subscript indicates the corresponding sentiment polarity.

| Models \ Datasets | Lap14 | | Rest14 | |
|---|---|---|---|---|
| | Acc. | F1 | Acc. | F1 |
| BERT | 75.80 | 71.67 | 82.59 | 74.10 |
| *BERT + augmentation* | | | | |
| + Prompt 1 | **77.74** | **73.11** | **84.29** | **76.07** |
| + Prompt 2 | 77.46 | 72.60 | 83.78 | 75.60 |
| + Prompt 3 | 77.50 | 72.94 | 84.08 | 75.72 |

Table 8: Performance comparison of BERT+augmentation using different prompts. The best scores are in bold.

methods have successfully reduced errors arising from statistical spurious correlations. Thus, we present a simple error analysis in Table 7. Taking the first sample as an example, BERT makes an incorrect prediction possibly because of focusing on the word *"friendly"* only. We suspect this is because, in the training samples which contain the word *"friendly"*, 94.6% of aspect sentiments are *"positive"*, *i.e.*, statistically spurious correlation. Moreover, we find that the proposed method also makes a few wrong predictions, especially when the true label is neutral, as shown in the second sample. The potential reason may be that when the label is neutral, the language model generates explanations with slight sentiments due to its own bias, which would mislead the model.