# OpenReview forum: "Reducing Spurious Correlations in Aspect-based Sentiment Analysis with Explanation from Large Language Models"
_EMNLP/2023/Conference — EMNLP 2023 Findings_

### Official Review · Reviewer_BR1K · 2023-07-25

**Soundness:** 4

**Excitement:**

3: Ambivalent: It has merits (e.g., it reports state-of-the-art results, the idea is nice), but there are key weaknesses (e.g., it describes incremental work), and it can significantly benefit from another round of revision. However, I won't object to accepting it if my co-reviewers champion it.

**Paper Topic And Main Contributions:**

The paper proposes a 2 step process to address spurious correlations in aspect-based sentiment analysis. Step 1: generate an explanation for a (sentence, aspect, polarity) triple using an LLM. Step 2: either augment the training data of a classifier using these explanations or using the explanations in a distillation setup.

**Questions For The Authors:**

- Q1: One of the assumptions in this paper is that spurious correlations are an actual problem. E.g., Sec. 3.3 states this as a potential problem. Did you run any analysis how severe of an issue this actually is? How often does it occur for a given model on a given task? Which models are more/less prone to this?
- Q2: L215: The definition assumes there's only one aspect per sentence. Are you considering sentences with multiple aspects?
- Q3: Sec. 3.3: Prompt-tuning is a difficult task. How did you come up with this template? Did you try others? Consider adding more information into the appendix.
- Q4: Sec 3: Did you run any analysis to probe for the quality of the automatically-generated explanations? I'd be curious if the explanations are accurate.
- Q5: Sec. 3.4 augmentation-based method: This approach supplies more training data to the classifier (according to Fig 2), which gives it an advantage compared to the baseline model. Have you compared the performance when keeping the number of training examples equal (e.g., by using 50% original, 50% explanations)? Without this we can't say for sure if improvements come from explanations or simply more training data.
- Q6: Eq. 2: I need some help understanding the Equations and the explaining text in L324-326. It could mean that the goal is to minimize the KL divergence between the predictions of the original example and the explanation for this example. In this case $g$ would be the student. Alternatively, it could mean that the goal is to minimize the KL divergence from the teacher's predictions and the student's predictions (which is the typical knowledge distillation process). In that case Eq. 2 would need to distinguish teacher and student and use $X'$ instead of $X$ or $X_hat$, e.g.,$KL(g_{teacher}(X'_i), g_{student}(X'_i)$. Similarly for Eq. 3. I may misinterpret something so I'd be grateful for any clarification.
- Q7: L412: Since we don't know what percentage of the data suffers from spurious correlation this claim is unjustified. The numbers simply show that more training data (augmentation) and distillation improves on the datasets. That in itself is nice, but there's (not necessarily) a relation to spurious correlations. On top of that we can't tell from the experiments if the given explanations are causing improvements or if its simply about adding more data in general.
- Q8: L420: Again, this statement is too strong. It may be a hypothesis, but not a fact that's shown by the experiments. It would help the reader to have a significance test to judge the difference in performance. E.g., Bert+PT vs GPT2 on Lag14 is < .5%, which is only 12 examples on this dataset.
- Q9: Tab 2: Does "+ distillation mean": "model + distillation" or "model + augmentation + distillation"?
- Q10: Sec 5, L456: Did you replace training examples with their augmented versions or add them to the existing set?
- Q11: Sec 5, L473: It would be useful to add 1 or 2 sentences how the adversarial setting works.
- Q12: L513: The interpretation here depends on my earlier question Q5. Does the augmentation model have twice the amount of data? If so it's not surprising that the model performs better and it's not clear if the improvement comes purely from more data or from the explanations themselves.

**Reasons To Accept:**

The proposed methods are intuitive (e.g., how to get an explanation and how to use them) and the many experiments show deep analysis and an attempt to study the problem from many different perspectives (e.g., text length, alternative text augmentation methods, robustness and generalization experiments).

**Reasons To Reject:**

- My main concern with the paper is that it's focussed around fixing spurious correlations but there's no analysis how much of a problem that actually is in any of the used datasets. Many of the experimental results are used as claims to show how spurious correlations are prevented but there's no proof that this is actually the case. Yes, performance of the models improve with the proposed methods (which in itself is a nice finding). However, we can't say that improvement are caused by less spurious correlations in the model; making contribution 1 (L141) invalid. In my view the paper needs more support for the claims or a rewrite away from spurious correlations.
- Some details are missing to fully judge the quality of the improvements and a simple baseline without adding additional training data is missing (see details below).

**Reproducibility:**

4: Could mostly reproduce the results, but there may be some variation because of sample variance or minor variations in their interpretation of the protocol or method.

**Reviewer Confidence:**

4: Quite sure. I tried to check the important points carefully. It's unlikely, though conceivable, that I missed something that should affect my ratings.

**Typos Grammar Style And Presentation Improvements:**

- Fig 2, step 1
	- The shadow under the text makes it difficult to read. I suggest deactivating it.
	- The turquoise text in the right-most box is pretty much unreadable. Please also consider using colors that are strong in contrast to also support readers that are visually impaired.
- L256: "unexplored" -> "explored"
- L317: nit: the footnote should go behind the period.
- Be careful with shrinking vertical spaces, e.g., Eq. 3 blends into the text below it.
- Tab 1: nit: align the numbers to the right
- Sec. 3.4, L304: I suggest to add a reference to Sec 5 to support this claim.
- L335: Careful with $\theta^T$, as this is standard mathematical notation for transposition. Maybe use $\theta$ for teacher and $\phi$ for student weights.

---

> ### Author Rebuttal · Authors · 2023-08-28
>
> ### **`Response to Reviewer BR1K`**
>
> Thanks very much for your valuable suggestions and comments. We sincerely appreciate your time in reading the paper, and our point-to-point responses to your comments are below.
>
> **[Comment 1]**
> *My main concern with the paper is that it's focused around fixing spurious correlations but there's no analysis how much of a problem that actually is in any of the used datasets. Many of the experimental results are used as claims to show how spurious correlations are prevented but there's no proof that this is actually the case. Yes, performance of the models improve with the proposed methods (which in itself is a nice finding). However, we can't say that improvement are caused by less spurious correlations in the model; making contribution 1 (L141) invalid. In my view the paper needs more support for the claims or a rewrite away from spurious correlations.*
>
> **[Response]**
> Thanks very much for your insightful comments and suggestions! In this work, spurious correlation (Zhang et al., 2022; Wang et al., 2022b) refers to the dependence of the model on certain words in the input text without a deeper understanding of the contextual semantics, which has a know-it-when-you-see-it character. **To the best of our knowledge, there are no relevant studies that analyze how much of the spurious correlation actually is in the used datasets. Existing related work only illustrates the presence of spurious correlations in datasets in the form of case studies.** Here, to answer your concerns and provide more support, we conduct a simple probe experiment on the Aspect Robustness Test Set (ARTS) (Xing et al., 2020). ARTS enriches the initial test sets from Lap14 and Rest14 by employing adversarial strategies (see some examples in Table below).
>
> |                     |                     **Sample**                    | **Aspect** | **Label** |
> |:---------------------|:-------------------------------------------------|:------------:|:-----------:|
> |   original sample   | Thanks for the fast shipment and great price.     |  shipment  |  positive |
> | adversarial sample | Thanks for the slow shipment and great price.     |  shipment  |  negative |
> | adversarial sample       | Thanks for the fast shipment and not great price. |  shipment  |  positive |
>
>
> Here, *if a model predicts the same sentiment labels for an original sample as well as its adversarial samples (their true labels are different), we will assume that there is a spurious correlation between certain contextual words (such as *"Thanks"*) and the predicted label (such as *"positive"*) for this original sample.* In other words, the predicted label does not change with the contextual semantics because the model only focuses on certain words.-- Based on this assumption, we count the percentage of original samples in the test set that contain spurious correlations (the lower the better). *It is worth reminding that this percentage is not the actual percentage of spurious correlations in the dataset, which is only an estimate under this assumption.* According to this table, we can see that the spurious correlations exist in the Lap14 and Rest14 datasets. Moreover, we can observe that the proposed methods reduce the percentage of original samples containing spurious correlations. This may suggest that the generated explanations can alleviate the spurious correlations problem in the ABSA task.
>
> |               | **Lap14** | **Rest14** |
> |---------------|:---------:|------------|
> | BERT          |   14.8%   |    14.5%   |
> | +augmentation |   12.7%   |    11.6%   |
> | +distillation |   13.0%   |    11.4%   |
> | AEN           |   16.3%   |    16.4%   |
> | +augmentation |   15.2%   |    14.8%   |
> | +distillation |   15.3%   |    15.0%   |
> | RGAT          |   12.4%   |    12.5%   |
> | +augmentation |   11.1%   |    11.6%   |
> | +distillation |   10.8%   |    11.0%   |
>
> >[1] Zhang, Michael, et al. "Correct-N-Contrast: a Contrastive Approach for Improving Robustness to Spurious Correlations." International Conference on Machine Learning. PMLR, 2022.
>
> >[2] Wang, Tianlu, et al. "Identifying and Mitigating Spurious Correlations for Improving Robustness in NLP Models." Findings of the NAACL. 2022.
>
> >[3] Xing, Xiaoyu, et al. "Tasty Burgers, Soggy Fries: Probing Aspect Robustness in Aspect-Based Sentiment Analysis." Proceedings of EMNLP. 2020.
>
> ---
>
> **[Comment 2]**
> *Some details are missing to fully judge the quality of the improvements and a simple baseline without adding additional training data is missing (see details below).*
>
> **[Response]**
> Thank you for this constructive comment. Since Question 5 is the detailed version of this comment, we will answer your concern in Answer 5.
>
> ---
> ---
> ---
>
> **[Q1]:**  *One of the assumptions in this paper is that spurious correlations are an actual problem. E.g., Sec. 3.3 states this as a potential problem. Did you run any analysis how severe of an issue this actually is? How often does it occur for a given model on a given task? Which models are more/less prone to this?*
>
> **[A1]:** Thank you for these questions to improve our manuscript. We answer these questions with the table in the response to Comment 1 and Table 4.
>
> For the first question, spurious correlations seriously hurt performance (see Table 4). This is because although they are a relatively small percentage (see the table above), they can prevent the model from achieving satisfactory results on the ARTS dataset. These results just amplify the harmful nature of this issue.
>
> For the second question, for a given model on a given task, how often it occurs depends on the training dataset. Suppose that if aspect sentiments are all *"positive"* in the training samples in which the word *"fast"* is contained, for a test sample containing *"fast"*, the trained model will predict *"positive"* with high probability regardless of the contextual semantics.
>
> For the third question, we can observe based on the table above that the attention\-based models (or the graph\-based models) may be more (or less) prone to this issue. We conjecture that the attention\-based models are more prone to concentrate on certain sentiment\-related words rather than understanding the entire contextual semantics. This may easily lead to statistically spurious correlations. In contrast, the graph\-based models can better understand the whole contextual semantics through the propagation of information between nodes.
>
> ---
>
> **[Q2]:**  *L215: The definition assumes there's only one aspect per sentence. Are you considering sentences with multiple aspects?*
>
> **[A2]:** Sorry for the unclear expression. For sentences with multiple aspects, we treat other non-targeted aspects as normal context tokens when focusing on the target aspect. In other words, a sentence will be processed multiple times, which is equal to the number of aspects it contains.
>
> Here, taking *"Great food but the service was dreadful!"* as an example for better illustration, we process it twice. For the first process, the model receives the aspect and sentiment as *"food"* and *"positive"*. For the second process, the model takes *"service"* and *"negative"* as the aspect and sentiment.
>
> ---
>
>
> **[Q3]:**  *Sec. 3.3: Prompt-tuning is a difficult task. How did you come up with this template? Did you try others? Consider adding more information into the appendix.*
>
> **[A3]:** Thanks for your advice. This prompt template is designed based on two points: (1) including a task description (i.e., *"explain the sentiment expressed by aspect term"*), which is used to trigger the LLM's ability to understand the task; (2) including the output limitation (i.e., *"Limit to forty words. Copying adjectives from the original sentence is not allowed."*), which is used as an output guide and prevents LLM from rephrasing the sentiment expression.
>
> To further answer your concern, we try other well-designed templates here and perform comparison experiments. These templates are described in detail below:
> - **Prompt 1** (used in this paper): In the following sentence $X$, explain why the sentiment expressed by aspect term $a$ is $y$. Limit to forty words. Copying adjectives from the original sentence is not allowed.
> - **Prompt 2**: In the following sentence $X$, explain why the sentiment expressed by aspect term $a$ is $y$.
> - **Prompt 3**: In the following sentence $X$, the sentiment expressed by aspect term $a$ is $y$. Based on this prompt, explain its reasoning. Limit to forty words. Copying adjectives from the original sentence is not allowed.
>
> Here, for simplicity, we choose BERT as the baseline model and BERT+augmentation as our method. The table below presents the experimental results. From this table, we can find that although the template used in this paper yields the best results, there is not much difference between the scores of the different templates (see Prompt1 vs. Prompt3). Moreover, we can observe that imposing an output constraint in the template favors the performance (see Prompt1 vs. Prompt2).
>
> |               |         | **Lap14** |        | **Rest14** |        |
> |---------------|---------|:---------:|:------:|:----------:|:------:|
> |               |         |  **Acc.** | **F1** |  **Acc.**  | **F1** |
> | BERT          |         |   75.80   |  71.67 |    82.59   |  74.10 |
> | +augmentation | +Prompt1 |   **77.74**   |  **73.11** |    **84.29**   |  **76.07** |
> | +augmentation | +Prompt2 |   77.46   |  72.60 |    83.78   |  75.60 |
> | +augmentation | +Prompt3 |   77.50   |  72.94 |    84.08   |  75.72 |
>
>
> ---
>
> **[Q4]:**  *Sec 3: Did you run any analysis to probe for the quality of the automatically\-generated explanations? I'd be curious if the explanations are accurate.*
>
> **[A4]:** Many thanks for this interesting question. In this work, we only randomly selected nearly fifty explanations for hand-checking. We find that the quality of the automatically generated explanations is satisfactory and accurate (see some examples in Table 7 in the Appendix). However, we did not quantitatively evaluate the quality of the explanations, which is an oversight on our part. In the revision version, we will try to find an approach to quantitatively probe the accuracy of the automatically generated explanations.
>
> ---
>
> **[Q5]:**  *Sec. 3.4 Augmentation\-based method: This approach supplies more training data to the classifier (according to Fig 2), which gives it an advantage compared to the baseline model. Have you compared the performance when keeping the number of training examples equal (e.g., by using 50% original, 50% explanations)? Without this we can't say for sure if improvements come from explanations or simply more training data.*
>
> **[A5]:** We are very grateful for this insightful question. To address your concern, we perform comparative experiments under different settings in which the proportions of original examples and explanations change but the totals remain consistent. The table below gives detailed experimental results. By comparing results when keeping the number of training examples equal, we can observe that the explanations can improve performance. However, we find that the magnitude of performance improvement does not increase as the proportions of explanations increase. We conjecture that the explanations have limited performance improvement under keeping the total number of training samples equal.
>
> Moreover, we can observe from Table 3 that even when doubling the number of training examples (e.g., by using 100% original, 100% explanations or augmentations), the proposed method obtains better results when compared to other data augmentation models. This suggests that the explanations used in this paper are indeed effective and can contribute to performance improvement.
>
> |          |              | **Lap14** |           | **Rest14** |           |
> |----------|--------------|:---------:|:---------:|:----------:|:---------:|
> | original | explanations |  **Acc.** |   **F1**  |  **Acc.**  |   **F1**  |
> |   100%   |      0%      |   75.80   |   71.67   |    82.59   |   74.10   |
> |    70%   |      30%     |   75.93   |   71.89   |    82.71   |   74.45   |
> |    50%   |      50%     |   76.45   |   72.53   |    83.60   |   75.03   |
> |    30%   |      70%     |   76.23   |   72.36   |    83.43   |   75.11   |
> |   100%   |     100%     | **77.74** | **73.11** |  **84.29** | **76.07** |
>
> ---
>
> **[Q6]:**  *Eq. 2: I need some help understanding the Equations and the explaining text in L324\-326. It could mean that the goal is to minimize the KL divergence between the predictions of the original example and the explanation for this example. In this case $g$ would be the student. Alternatively, it could mean that the goal is to minimize the KL divergence from the teacher's predictions and the student's predictions (which is the typical knowledge distillation process). In that case Eq. 2 would need to distinguish teacher and student and use $X'$ instead of $X$ or $\hat{X}$, e.g.,$KL(g_{teacher}(X'_i),g\_{student}(X'_i))$. Similarly for Eq. 3. I may misinterpret something so I'd be grateful for any clarification.*
>
>
> **[A6]:** Thanks for your meticulous review and for raising questions about the equations and associated explaining text in lines 324-326. You are correct in your understanding. Our goal is indeed to minimize the KL divergence between the predictions of the original sample and the explanation for this sample. The equations we provided for this purpose aim to mathematically describe this process. In light of your feedback, we will revise these two equations and make the necessary text modifications to differentiate clearly between the teacher and student models.
>
> Thank you again for pointing out these ambiguities!
>
>
>
> ---
> **[Q7]:**  *L412: Since we don't know what percentage of the data suffers from spurious correlation this claim is unjustified. The numbers simply show that more training data (augmentation) and distillation improves on the datasets. That in itself is nice, but there's (not necessarily) a relation to spurious correlations. On top of that we can't tell from the experiments if the given explanations are causing improvements or if its simply about adding more data in general.*
>
> **[A7]:** Thanks very much for your insightful comments! Here, we address your concerns from two perspectives: (1) According to the table in the response to Comment 1, the percentage of the data that suffers from spurious correlation decreases after applying the explanations. This shows that the proposed explanations can effectively mitigate spurious correlations; (2) According to the Answer to the corresponding Question5, we can know that the explanations can bring performance improvement when keeping the number of training examples equal. In the revision, we will supplement a detailed description so that readers can better understand our work.
>
>
> ---
>
> **[Q8]:**  *L420: Again, this statement is too strong. It may be a hypothesis, but not a fact that's shown by the experiments. It would help the reader to have a significance test to judge the difference in performance.*
>
> **[A8]:** Thanks for your useful advice. In the next version, we will re-examine the paper and modify inappropriate statements. Here, in response to your suggestion, we add a significance test in the table below to judge the difference in performance described by line 420. The symbol \* denotes that the score is significantly better than that of GPT2 at a significance level of $p < 0.05$.    We can see that although the number of parameters in GPT2 is slightly larger than that in BERT+PT, BERT+PT is overall better than GPT2. The potential reason may be that understanding the context sufficiently related to the given aspect is difficult when using a text generation paradigm to solve ABSA.
>
> |         | **Lap14** |        | **Rest14** |        | **Rest15** |        | **Rest16** |        | **MAMS** |        |
> |:-------:|:---------:|:------:|:----------:|:------:|:----------:|:------:|:----------:|:------:|:--------:|:------:|
> |         |  **Acc.** | **F1** |  **Acc.**  | **F1** |  **Acc.**  | **F1** |  **Acc.**  | **F1** | **Acc.** | **F1** |
> | BERT+PT |   77.59   |  $73.14^*$ |    84.16   |  76.62 |    $82.83^*$   |  $65.81^*$ |    $91.73^*$   |  $73.93^*$ |   83.48  |  $83.10^*$ |
> |   GPT2  |   76.04   |  72.56 |    84.11   |  77.02 |    81.06   |  64.90 |    89.96   |  71.90 |   82.86  |  81.61 |
>
> ---
> **[Q9]:**  *Tab 2: Does "+ distillation mean": "model + distillation" or "model + augmentation + distillation"?*
>
> **[A9]:** Sorry for the unclear description. In Table 2, *"+ distillation"* refers to *"model + distillation"*.
>
> ---
>
> **[Q10]:** *Sec 5, L456: Did you replace training examples with their augmented versions or add them to the existing set?*
>
> **[A10]:** We apologize for not going into detail. In line 456, we add these augmented versions directly to the existing set. The size of the training set will be doubled after this operation.
>
> ---
>
> **[Q11]:** *Sec 5, L473: It would be useful to add 1 or 2 sentences how the adversarial setting works.*
>
> **[A11]:** Thanks for your useful suggestions. In line 473, to show that our methods can reduce spurious correlations (i.e., robustly understand the contextual semantics in which the aspect sentiment resides rather than focusing on certain words), we conduct experiments on the Aspect Robustness Test Set (ARTS). This dataset simulates some adversarial scenarios. Specifically, ARTS extends the original test samples of Lap14 and Rest14 using three adversarial settings: (1) reversing the original sentiment of the targeted aspect; (2) reversing the sentiment of the non\-targeted aspects, and (3) generating more non-targeted aspects with opposite sentiment polarities from the targeted aspect. Thus, this dataset is able to test whether the model can robustly capture the aspect\-relevant contextual semantics to distinguish the sentiment towards the targeted aspect from the input text.
>
>
>
> ---
>
> **[Q12]:**  *L513: The interpretation here depends on my earlier question Q5. Does the augmentation model have twice the amount of data? If so it's not surprising that the model performs better and it's not clear if the improvement comes purely from more data or from the explanations themselves.*
>
> **[A12]:** We apologize for the unclear description. In line 513, the augmentation model has twice the amount of data. However, as described in Anaswer5, the explanations do bring performance improvement. Thus, we could roughly conclude that the performance improvement of line 513 comes not only from more data but also from the explanations themselves.
>
> ---
>
> **`Typos Grammar Style and Presentation Improvements`**
>
> Thanks for your detailed reading! In order to improve the quality of the presentation, we will fix the typos and bugs you pointed out in the revision following your suggestions.
>
>
>
> ---
>
> We sincerely thank you for your positive feedback on our paper! If you have any questions or suggestions, please let us know!
>
> Best,
>
> Authors

---

### Official Review · Reviewer_vdgb · 2023-08-02

**Soundness:** 4

**Excitement:**

4: Strong: This paper deepens the understanding of some phenomenon or lowers the barriers to an existing research direction.

**Paper Topic And Main Contributions:**

The paper presents a novel data augmentation strategy for ABSA (targeted sentiment classification) based on the use of large language models (LLM).

I like the motivation of augmenting training data for ABSA for mitigating the spurious statistical correlation of the presence of some words with a specific sentiment label. Likewise, I see that the evaluation and analysis of the method is right. However, I miss an error analysis in order to see if the spurious correlations have disappeared.

To sum up, I think that it is a good paper with the lack of an error analysis.



**Questions For The Authors:**

- Does the method mitigate all the spurious correlations?
- Are there any new spurious correlations?

**Reasons To Accept:**

- The novelty of using LLM for augmenting data for ABSA.
- The strong results of the two training approaches desinged.

**Reasons To Reject:**

- The lack of an error analysis to corroborate the claim.

**Reproducibility:**

3: Could reproduce the results with some difficulty. The settings of parameters are underspecified or subjectively determined; the training/evaluation data are not widely available.

**Reviewer Confidence:**

5: Positive that my evaluation is correct. I read the paper very carefully and I am very familiar with related work.

---

> ### Author Rebuttal · Authors · 2023-08-28
>
> ### **`Response to Reviewer vdgb`**
>
> Thanks very much for your valuable suggestions and comments. We sincerely appreciate your time in reading the paper, and our point-to-point responses to your comments are below.
>
> **[Comment 1]**
> *The lack of an error analysis to corroborate the claim.*
>
> **[Response]**
> Thank you for the comment to improve our manuscript. An error analysis can provide readers with a deeper understanding of whether our methods have successfully reduced errors arising from statistical spurious correlations. In the next version, we will supplement the error analysis to better understand the claim of this paper.
>
> Here, in response to your comment, we present a simple error analysis in the table below. In Table, aspect terms are marked in parentheses, and the subscript indicates the corresponding sentiment polarity. Taking the first sample as an example, BERT makes an incorrect prediction possibly because of focusing on the word *"friendly"* only. We suspect this is because in the training samples which contain the word *"friendly"*, 94.6% of aspect sentiments are *"positive"*, i.e., statistically spurious correlation. In contrast, the proposed method can understand the contextual semantics rather than focusing on certain words when modeling the sentiment feature of the aspect. Moreover, we find that the proposed method also makes a few wrong predictions, especially when the true label is neutral, as shown in the second sample. The potential reason may be that when the label is neutral, the language model generates explanations with slight sentiments due to its own bias, which would mislead the model.
>
> |                                                  **Sample**                                                  | **BERT** | **BERT+augmentation** |
> |:------------------------------------------------------------------------------------------------------------|:----------:|:-----------------------:|
> | The $[staff]_{negative}$ should be a bit more friendly.                                                           | positive |        negative       |
> | I needed a laptop with big $[storage]_{neutral}$,  a nice screen and fast so I can Photoshop without any problem. | positive |        positive       |
>
> ---
>
> **[Comment 2]**
> *Does the method mitigate all the spurious correlations?*
>
> **[Response]**
> Thank you very much for the question. We regret to answer that mitigating all spurious correlations is absolute and difficult as well as somewhat impractical. In this paper, we aim to reduce these spurious correlations as significantly as possible and improve the model's generalization capabilities, as shown by various experiments.
>
>
> ---
>
> **[Comment 3]**
> *Are there any new spurious correlations?*
>
> **[Response]**
> Thank you for this constructive question. Indeed, detecting whether a few new spurious correlations are introduced is required after applying LLM to generate explanations. In this work, we only manually examine the quality of the random sampling explanations (see examples in Table 7). We find that these explanations are richly expressive (e.g., *"not reasonable or affordable"*). Therefore, we consider that they do not introduce new spurious correlations. Nevertheless, to answer this question accurately, we will subsequently find a way to do a quantitative analysis in the next version.
>
> ---
>
> We sincerely thank you for your positive feedback on our paper! If you have any questions or suggestions, please let us know!
>
> Best,
>
> Authors

---

### Official Review · Reviewer_NFXw · 2023-08-04

**Soundness:** 3

**Excitement:**

2: Mediocre: This paper makes marginal contributions (vs non-contemporaneous work), so I would rather not see it in the conference.

**Paper Topic And Main Contributions:**

The paper presents a 2-step framework that leverages explanations generated by large language models to augment existing aspect-based sentiment analysis (ABSA) dataset and so forth improve the performance on such task. The framework contains a prompt-based explanation generation and two simple yet effective explanation utilization strategies.

**Reasons To Accept:**

The framework the study proposed achieved a general improvement on the ABSA task against multiple benchmark datasets.

The study includes solid experiments comparing various approaches on the specific task.

**Reasons To Reject:**

Compared to other data augmentation baselines, the improvement achieved is relatively limited, considering that the framework could be resource-wise demanding since it requires loads of language model generation and training.

Relying solely on unexamined and potentially biased language model generations require more caution, since language models could not always provide accurate and comprehensive explanations. I would suggest conduct a closer investigation on LLM generated explanations and subsequently apply necessary quality control measures.

Concerning reproducibility, the snapshot version and parameters of the LLMs used (GPT3.5 and MOSS) are not unspecified.

**Reproducibility:**

3: Could reproduce the results with some difficulty. The settings of parameters are underspecified or subjectively determined; the training/evaluation data are not widely available.

**Reviewer Confidence:**

4: Quite sure. I tried to check the important points carefully. It's unlikely, though conceivable, that I missed something that should affect my ratings.

---

> ### Author Rebuttal · Authors · 2023-08-28
>
> ### **`Response to Reviewer NFXw`**
>
> Thanks very much for your valuable suggestions and comments. We sincerely appreciate your time reading the paper, and our point-to-point responses to your comments are below.
>
> **[Comment 1]**
> *Compared to other data augmentation baselines, the improvement achieved is relatively limited, considering that the framework could be resource-wise demanding since it requires loads of language model generation and training.*
>
> **[Response1]**
> Thanks very much for your insightful comment! In this paper, we leverage explanations from LLMs to reduce spurious correlations in ABSA. Compared to other augmentation instances in which modified strategies are applied to the input text, these explanations are more readable. Also, they are more effective because they promote understanding of the contextual semantics when reasoning about the sentiments of the aspects rather than focusing only on certain words. Based on the experiments in Table 3, we can see that most of our scores are significantly better than those of other data augmentation baselines at a significance level of $p$ < 0.05.
>
> In addition, another advantage of the proposed methods over other data augmentation baselines is the ability to improve the model's robustness. To test this point, we conduct experiments on the Aspect Robustness Test Set (Xing et al., 2020) and report results in the table below. We can find that the proposed methods (i.e., +augmentation and +distillation) yield a remarkable improvement compared to two popular augmentation baselines (i.e., +ADD and +DELETE).
>
> Finally, in order to clearly understand resource-wise demands, we would like to clarify that the proposed methods only load the language model for inference without involving training.
>
>
> |               |          **Lap14**         |  |  **Rest14**      ||
> |:-------------|:-----------:|:-----------:|:-----------:|:-----------:|
> |               | **Acc.**  | **F1**    | **Acc.**  | **F1**    |
> | BERT          |   60.58   |   56.41   |   60.74   |   52.02   |
> |          +ADD |   62.39   |   58.42   |   68.56   |   59.40   |
> |       +DELETE |   63.18   |   58.06   |   67.79   |   58.54   |
> | +augmentation | **66.82** | **62.28** |   74.13   | **66.00** |
> | +distillation |   66.09   |   61.12   | **74.22** |   65.47   |
>
> ---
>
>
> **[Comment 2]**
> *Relying solely on unexamined and potentially biased language model generations requires more caution since language models cannot always provide accurate and comprehensive explanations. I would suggest conduct a closer investigation on LLM generated explanations and subsequently apply necessary quality control measures.*
>
> **[Response]**
> Thank you for this constructive suggestion. Indeed, language models may generate less accurate or even irrelevant explanations for the sentiment of the aspect. Such explanations may have negative effects if they exist and are involved in training. In this work, we only randomly selected nearly fifty explanations for hand-checking. We find that the quality of the generated explanations is accurate and comprehensive (see examples in Table 7 in the Appendix). In response to your suggestion, we will purposefully design a module in the next version to determine the accuracy of the explanations generated by LLM and then take some control measures based on the accuracy.
>
> ---
>
>
> **[Comment 3]**
> *Concerning reproducibility, the snapshot version and parameters of the LLMs used (GPT3.5 and MOSS) are not unspecified.*
>
> **[Response]**
> Very sorry for the lack of detail. The snapshot version and parameters of the GPT3.5 are [text-davinci-003](https://platform.openai.com/docs/models/gpt-3-5) and 175B, respectively. The snapshot version and parameters of the MOSS are [MOSS-moon-003-sft](https://github.com/OpenLMLab/MOSS) and 16B, respectively.
>
> ---
>
>
> We sincerely thank you for your positive feedback on our paper! If you have any questions or suggestions, please let us know!
>
> Best,
>
> Authors

---

### Meta-Review · Area_Chair_qxN8 · 2023-09-26

**Recommendation:** 3

**Metareview:**

This paper introduces a 2-step framework using large language models (LLMs) to augment aspect-based sentiment analysis (ABSA) datasets, aiming to improve performance. The framework involves prompt-based explanation generation and two explanation utilization strategies.

Reasons to Accept:
- The proposed framework demonstrates a significant overall improvement in aspect-based sentiment analysis (ABSA) across multiple benchmark datasets.
- The study features robust experiments that compare various approaches on the specific task, providing strong empirical evidence.
-The novelty of leveraging large language models (LLMs) for data augmentation in ABSA
-The study presents two effective training approaches, enhancing the overall quality of the proposed methods.
-The methods are intuitive, and the extensive experiments offer a deep analysis from various perspectives, including text length, alternative text augmentation methods, robustness, and generalization.

Reasons to Reject:
- The observed improvement achieved with the proposed framework is relatively limited when compared to other data augmentation baselines. Given the resource-intensive nature of the framework, this raises concerns about its cost-effectiveness.
- Relying solely on unexamined and potentially biased language model-generated explanations may not be a reliable approach, as language models may not consistently provide accurate or comprehensive explanations. A closer investigation of LLM-generated explanations and the application of quality control measures are needed.
- The paper lacks transparency regarding the snapshot version and parameters of the LLMs used (GPT-3.5 and MOSS), which affects reproducibility.
- The absence of an error analysis to validate the claim is a significant drawback.

---

### Decision · Program_Chairs · 2023-10-07

**Decision:**

Accept-Findings

**Comment:**

This paper introduces a 2-step framework using large language models (LLMs) to augment aspect-based sentiment analysis (ABSA) datasets, aiming to improve performance. The framework involves prompt-based explanation generation and two explanation utilization strategies.

Reasons to Accept:
- The proposed framework demonstrates a significant overall improvement in aspect-based sentiment analysis (ABSA) across multiple benchmark datasets.
- The study features robust experiments that compare various approaches on the specific task, providing strong empirical evidence.
-The novelty of leveraging large language models (LLMs) for data augmentation in ABSA
-The study presents two effective training approaches, enhancing the overall quality of the proposed methods.
-The methods are intuitive, and the extensive experiments offer a deep analysis from various perspectives, including text length, alternative text augmentation methods, robustness, and generalization.

Reasons to Reject:
- The observed improvement achieved with the proposed framework is relatively limited when compared to other data augmentation baselines. Given the resource-intensive nature of the framework, this raises concerns about its cost-effectiveness.
- Relying solely on unexamined and potentially biased language model-generated explanations may not be a reliable approach, as language models may not consistently provide accurate or comprehensive explanations. A closer investigation of LLM-generated explanations and the application of quality control measures are needed.
- The paper lacks transparency regarding the snapshot version and parameters of the LLMs used (GPT-3.5 and MOSS), which affects reproducibility.
- The absence of an error analysis to validate the claim is a significant drawback.